# Evaluation of a Hybrid Moving Bed Biofilm Membrane Bioreactor and a Direct Contact Membrane Distillation System for Purification of Industrial Wastewater

**DOI:** 10.3390/membranes13010016

**Published:** 2022-12-22

**Authors:** Mamdouh S. Alharthi, Omar Bamaga, Hani Abulkhair, Husam Organji, Amer Shaiban, Francesca Macedonio, Alessandra Criscuoli, Enrico Drioli, Zhaohui Wang, Zhaoliang Cui, Wanqin Jin, Mohammed Albeirutty

**Affiliations:** 1Department of Mechanical Engineering, King Abdulaziz University, P.O. Box 80200, Jeddah 21589, Saudi Arabia; 2Center of Excellence in Desalination Technology, King Abdulaziz University, P.O. Box 80200, Jeddah 21589, Saudi Arabia; 3Institute on Membrane Technology, National Research Council of Italy (CNR-ITM), Via P. Bucci 17/C, 87036 Rende, Italy; 4State Key Laboratory of Materials-Oriented Chemical Engineering, College of Chemical Engineering, Nanjing Tech University, Nanjing 211816, China

**Keywords:** industrial wastewater, moving bed biofilm reactor, membrane bioreactor, membrane distillation, hybrid process

## Abstract

Integrated wastewater treatment processes are accepted as the best option for sustainable and unrestricted onsite water reuse. In this study, moving bed biofilm reactor (MBBR), membrane bioreactor (MBR), and direct contact membrane distillation (DCMD) treatment steps were integrated successively to obtain the combined advantages of these processes for industrial wastewater treatment. The MBBR step acts as the first step in the biological treatment and also mitigates foulant load on the MBR. Similarly, MBR acts as the second step in the biological treatment and serves as a pretreatment prior to the DCMD step. The latter acts as a final treatment to produce high-quality water. A laboratory scale integrated MBBR/MBR/DCMD experimental system was used for assessing the treatment efficiency of primary treated (PTIWW) and secondary treated (STIWW) industrial wastewater in terms of permeate water flux, effluent quality, and membrane fouling. The removal efficiency of total dissolved solids (TDS) and effluent permeate flux of the three-step process (MBBR/MBR/DCMD) were better than the two-step (MBR/DCMD) process. In the three-step process, the average removal efficiency of TDS was 99.85% and 98.16% when treating STIWW and PTIWW, respectively. While in the case of the two-step process, the average removal efficiency of TDS was 93.83% when treating STIWW. Similar trends were observed for effluent permeate flux values which were found, in the case of the three-step process, 62.6% higher than the two-step process, when treating STIWW in both cases. Moreover, the comparison of the quality of the effluents obtained with the analysed configurations with that obtained by Jeddah Industrial Wastewater Treatment Plant proved the higher performance of the proposed membrane processes.

## 1. Introduction

The increased demand for freshwater has led to the increase in withdrawals of limited nonrenewable water resources, leading to water scarcity [1,2]. This scarcity has led to water and wastewater treatment innovations, and the need to follow better environmental practices. Stringent water quality standards have helped the evolution of advanced effluent treatment technologies, thereby preserving water quality [2,3]. The high water demand and the environmental threat added more pressure on managing and recycling water. Water reuse through bioreactors can be considered an additional source of environmental sustainability. Recent trends have made the industries either minimize production or recycle treated wastewater to reduce their effects and follow the concept of zero liquid discharge [3]. The effluents of the industrial wastewater treatment plants are characterized by the levels of their constituents, such as biological oxygen demand (BOD5), chemical oxygen demand (COD), total suspended solids (TSS), total dissolved solids (TDS), oil, and grease. Apart from these, industry-specific effluents can include various organic matters, toxins, heavy metals, phenols, dioxins, and furans. The degree of treatment required depends on the quality of effluent and its characteristics. Treatment can be done in aerobic, anaerobic, and anoxic conditions to obtain the desired quality of treated water [4,5]. 

The wastewater treatment process is realized by chemical, physical, physio-chemical, and biological or a combination of these methods. Biological methods include biofilters, trickling filters, biological contactors, Activated Sludge Process (ASP), Sequencing Batch Reactor, Membrane Bio Reactor (MBR), and Moving Bed Biofilm Reactor (MBBR). Chemical treatment includes aeration, chlorination, and disinfection methods. Physical treatment includes screening, sedimentation, filtration, and flotation. Physio-chemical processes include coagulation and flocculation. The advanced treatment includes carbon adsorption, absorption, stripping, ion exchange, reverse osmosis, and disinfection. Hybrid treatment plants include a combination of these processes for effective treatment [6]. The selection of the suitable combination depends on the wastewater’s characteristics and on the required quality of the obtained effluent.

In fact, a wastewater treatment process typically consists of four steps, namely primary treatment, secondary treatment, tertiary treatment, and advanced treatment. The primary treatment includes screening, neutralization, sedimentation, and flotation processes. It is meant to remove contaminants such as debris, grit, sand, etc. Secondary treatment includes aerobic and anaerobic treatments with sedimentation, and its function is to remove organic contaminants and ammonia. Tertiary treatment includes adsorption, precipitation, and disinfection; it is meant to remove nutrients and pathogens [7].

Membrane bioreactors (MBR) are used primarily in wastewater treatment. MBR involves the use of suspended mixed microbial cultures. MBR technology combines biological processes, such as ASP, with membrane filtration. The most common configuration is called a submerged membrane bioreactor [8]. Membrane bioreactors are configured by the type of separation they are designed for. The separation is carried out by either pressure-driven membranes in side-stream MBR’s or vacuum-driven membranes submerged in the reactor. In the side-stream MBRs, the wastewater is pumped through the membrane and returned to the bioreactor for further treatment; whereas, vacuum-driven membranes are submerged in the bioreactor [9]. The major advantage of MBR over conventional systems is its smaller carbon footprint [10]. MBR is up to one-third of the size of conventional ASP systems with the same treatment capacity. Moreover, low sludge production and higher-quality degraded sludge are produced in the MBR system. This advantage of MBR contributes to the better competitiveness of MBRs compared to ASPs [9]. The disadvantages of MBRs are their higher oxygen demand (requiring higher energy input when compared to conventional systems), and the fouling of the membranes (requiring constant monitoring and maintenance) [9].

The moving bed biofilm reactor (MBBR) was developed in Norway between the late 1980s and early 1990s by Odegaard H. et al., 1994 [11]. The MBBR process involves utilizing the entire volume of the reactor space for biomass growth. The process uses carriers that move freely in the reactor, acting as a biomass growth medium. The carriers are retained in the reactor by a sieve arrangement at the outlet of the reactor. MBBR can be used under aerobic, anaerobic, and anoxic conditions. For effective treatment, the biofilm carriers are required to be in motion; under the aerobic condition, the movement of carriers is caused by air movement, whereas in anaerobic and anoxic conditions, a mixer is used for agitation of the carriers. The carriers are made from high-density polyethylene (PEHD) with a density of 0.96 g/cm^3^. The biomass grows on the surface of these carriers, and due to the shape of the carriers, they provide a higher surface area for the growth of biomass. The ideal growth of biomass is thin and evenly distributed along the surface area of the carrier, for which turbulence is a crucial factor. Turbulence provides movement of the carriers and maintains a thin biomass layer through the shearing force [12]. Sohail et al. (2020) proposed the integration of MBBR with MBR for the mitigation of MBR membrane fouling [13]. The MBBR has a superior performance in reducing the concentration of suspended solids due to the high biodegradation rate of the organic matter facilitated by the biofilm carriers. They have concluded that the MBBMR was a superior option with respect to effluent rate, operation time, and sludge mass compared to stand alone MBR, MBBR, or ASP.

Membrane Distillation (MD) is a water desalination technology comprised of a hydrophobic membrane that only allows volatile components (and, therefore water vapor) to pass through, not the liquid. MD operates on a temperature difference between the hot feed, which is in contact with the upstream side of the membrane, and the cold condensate, which is in contact with the downstream side of the membrane [14,15]. The configuration can operate at relatively low temperatures of the feed, and has an excellent salts and pollutants rejection efficiency (close to 100%). However, MD is not as energy efficient as reverse osmosis (RO) systems. Jeong et al. [16] conducted a feasibility study of the MD process for the treatment of wastewater from sewage treatment plants for potable water reuse. It was found that MD achieved treated water quality levels as required for drinking purposes, most of the dissolved organic matter was rejected, and a few naturally found amino acids such as tyrosine passed through the membrane. All pharmaceuticals were removed in such a way that their concentrations were below the quantification limits; however, membrane fouling was found to be an issue [16].

The membrane distillation bioreactor (MDBR) combines the thermophilic biological process with the MD process [17]. A vapor-liquid interface is created in the MD process and passes the same through a hydrophobic membrane. The influent wastewater vaporizes at the vapor-liquid interface close to the surface of the hydrophobic membrane, and permeates under the effect of the vapor pressure gradient. Finally, it is condensed and the distillate is removed. Conventional MBR system uses either microfiltration or ultrafiltration microporous membranes to retain the biomass or mixed liquor within the reactor. In comparison, MDBR has an MD membrane to retain the same. MDBR membranes can be placed in a side-stream, or submerged in the bioreactor. MDBRs are operated at 50–60 °C to treat the influent wastewater. The average specific energy consumption (SEC) of the DCMD system was calculated at around 500 kWh/m^3^ [18] which is much higher than the SEC of the RO system. The high SEC of the MD process is attributed to the nature of the driving force of the process which implies the necessity of a temperature gradient across the membrane. For this reason, MD modules must have a heat sink on the permeate side to induce a temperature gradient. This makes MD work as a heat exchanger where most of the exergy of feed water is destructed and lost to the permeate side. In cases of the availability of waste heat on the treatment site, which is a common feature of many industries, the waste heat can be utilized suitably for MD operation [19]. Under such conditions, the MD system can produce freshwater without any high energy costs [20]. The MDBR systems, similar to MBR systems, are also prone to fouling. This can, however, be managed or controlled to a certain extent by way of bubbling and cleaning the membranes [21]. MDBR produces a higher quality of effluent than MBR and can be operated on the principle of using waste heat produced by specific industries [21]. Goh et al. [17] noted that the inclusion of biomass in the MDBR system could result in a decline in flux and bio-fouling. In their study, they successfully delayed wetting by 1.7–3.6 times by just lowering the retentate organic and nutrient concentration. Fast flux decline was due to the thermal and mass transfer resistance of the biofilm; however, the same cannot be controlled with periodic membrane cleaning and process optimization. It was concluded that MDBR can be used for the reclamation of industrial wastewater with low volatile organic content and can be feasible if access to waste heat is readily available.

Khaing et al. [22] conducted a study utilizing submerged MDBR for the treatment of petrochemical wastewater. Membranes were found to be thermally stable and could maintain the flux over 5.5 L/(m^2^ h) throughout the study period of 105 days; however, flux decline was found due to inorganic fouling of the membrane. Leyva-Diaz et al. [23] tested a hybrid MBBR-MBR system at two different scales of operation to analyze their effect on municipal wastewater treatment. The configurations were reliable for organic matter removal, with COD removal percentages of 90.97 ± 2.55% and 95.56 ± 2.01% for hybrid MBBR–MBRL and hybrid MBBR–MBRP, respectively. Trapani et al. [24] compared two pilot-scale MB-MBR and MBR systems by increasing salinity in feed wastewater. Pore fouling tendency was noted to be higher in the MBR system. It was concluded that the MB-MBR system performed better and had potential for treatment of high strength or industrial wastewater.

In this study, an MBBR combined with a UF membrane is introduced before a DCMD system to obtain the combined advantages of MBBR, MBR, and DCMD. The process was analysed as a hybrid system for industrial wastewater treatment. Two configurations of the hybrid moving bed biofilm membrane distillation bioreactor (MBBMDBR) system were assessed and compared for purification of primary and secondary treated industrial wastewater in terms of the system permeate water flux, quality, and membrane fouling.

## 2. Methodology

Three experiments were carried out in this study. The first and second experiments were devoted to evaluating the performance of a hybrid moving bed biofilm reactor (MBBR) combined with a UF membrane followed by a DCMD system for the treatment of primary and secondary wastewater, respectively. In the third experiment, the performance of the submerged MDBR system was assessed for the purification of secondary treated wastewater (Figure 1).

### 2.1. Materials

Feed water was collected at two locations from Jeddah Industrial Wastewater Treatment Plant: (a) primary treated water (F1) has undergone primary treatment of grit removal, oil and grease removal, and primary sedimentation, and (b) secondary treated water (F2) was collected from the secondary clarifier after activated sludge treatment. The physicochemical and biological analysis data of the feed wastewater is presented in Appendix A in Appendix A. F1 The influent feed water has a light-yellow tint to it. The major parameters analyzed were TDS (962 mg/L), pH (9.4), TSS (524 mg/L), Turbidity (40 NTU), and TOC (286 ppm). Major anions and cations were Chloride (342 ppm), Sodium (313 ppm), Sulphate (110 ppm), Potassium (31.9 ppm), Calcium (24 ppm), and Magnesium (7.3 ppm). Heavy metal content Aluminum (1.9 ppm), Iron (0.3 ppm), and Zinc (0.16 ppm).

F2 influent feed water has a greenish-yellow tint to it. Major parameters analyzed were TDS (818 mg/L), pH (9.4), TSS (90 mg/L), Turbidity (1.6 NTU), and TOC (11.7 ppm). Major anions and cations were Chloride (304 ppm), Sodium (286 ppm), Sulphate (36.8 ppm), Potassium (24.3 ppm), Calcium (16.6 ppm), and Magnesium (5.5 ppm). Heavy metal content in influent feed water-1 was found to be very low with the highest concentration being aluminum (0.4 ppm).

Hydrophilic PVDF ultrafiltration membranes were procured from Nanjing Tech University (NTU), Nanjing, China, and used to prepare experimental membrane modules for the moving bed biofilm membrane bioreactor MBBMR configuration (Figure 2a). Hydrophobic PVDF membranes (supplied by Econity, South Korea) were used for preparing submerged (Figure 2b) and side-stream (Figure 2c) membrane distillation modules. Specifications of UF modules and MD modules are provided in Table 1 and Table 2, respectively.

### 2.2. Experimental Setup Description

A bench scale integrated MBBMR-MD experimental setup was assembled to investigate the efficiency of integration of membrane bioreactor combined with UF membrane separation followed by membrane distillation. The experimental unit consists of standard process components and instrumentations of research quality mounted on a movable bench. A schematic diagram of the experimental setup as configuration-1 used in experiments 1 and 2 is shown in Figure 3, and configuration-2 used in experiment-3 is shown in Figure 4. An acrylic tank of thickness 10 mm with a removable cover (MBBMR Tank) of capacity 23.6 L has been used as a feed tank. Two ceramic air diffusers were provided at the bottom of the reactor to produce fine bubbles. A circular tank made of acrylic material with a volume of 8.5 L was used as an MD feed tank in configuration 1. The same tank was used as a membrane distillation bioreactor in configuration 2.

Polyurethane (PU) tubes were used to connect all tanks. A refrigerated bath has been used (Huber, model no MPC-K25, Germany) for cooling the MD permeate. Diaphragm pumps were used to transfer the influent from one tank to another. An air pump was used to maintain sufficient dissolved oxygen (DO) levels in the reactor. The transmembrane pressure of MBR was measured by using a pressure gauge. Needle valves were used at specific points to control the flow in the model reactor. A pressure transducer for the hot side membrane inlet is provided; similarly, a pressure transducer for the cold side membrane inlet is provided. Thermocouples measure the temperature at the hot side of the membrane (both inlet and outlet) as well as for the cold side (both inlet and outlet). A ball valve has been provided for flow control. A conductivity transmitter (Signet 9900 Transmitter) was provided to record the conductivity of treated water. A precision scale (GF-1200 precision scale from AandD Weighing) was used to measure the final treated water from the MD. All fittings and valves were made of stainless steel 316 L fittings. Finally, a data acquisition system (DAQ) was used to record the pressure, temperature, and mass of permeate on a timely basis.

## 3. Results and Discussion

The daily variations of the main quality parameters of water streams in terms of temperature, pH, TDS, BOD_5_, and TSS for each configuration are presented in this section. The readings of the parameters were taken at 4 locations in the setup, i.e., MBBMR tank (pH_0_, TDS_0_, BOD_5,0_, TSS_0)_, MBBMR filtrate (pH_T1_, TDS_T1_, BOD_5,T1_, TSS_T1_), MD feed tank (pH_T2_, TDS_T2_, BOD_5,T2_, TSS_T2_), and MD permeate (pH_p_, TDS_p_, BOD_5,p_, TSS_p_). Similarly, the MBBMR filtrate flux (F_0_) and MD permeate flux (J_p_) were recorded throughout the time of the tests. The operational duration of experiments is reported in the summary of results table (Table 3) which also shows the averages of performance parameters of the tested configurations.

### 3.1. Assessment of Hybrid MBBMR and DCMD Configuration Performance for Secondary Wastewater Treatment (Experiment 1) and Primary Wastewater Treatment (Experiment 2)

#### 3.1.1. TDS of MBBMR and MDBR Effluents

The treatment of the feed secondary wastewater was accomplished in two successive steps. The first step is the MBBMR treatment, and the second step is MD purification. As shown in Table 3, the average pH of the mixed liquor-suspended solids (MLSS) in the MBBMR tank was 7.9, and was observed to vary in the range of 7.7–8.2. This level of pH values is characteristic of industrial wastewater types. Although it was advised to maintain the pH of the activated sludge system at about 7 for the best growth of the microorganisms [25]. In this study, no pH control was adopted since the MBBR system can tolerate changes in temperature and pH. The pH of the MD permeate water was in the range of 6.0–6.57 range with an average value of 6.4, which is similar to the pH of the deionized water produced in the laboratory from tap water by a pure water RO unit. TDS readings were recorded and plotted for TDS_T2_ and TDS_P_ in Figure 5. For Experiment 1, the TDS_f_ values of the feed water ranged from 835–859 mg/L, however, an increase in TDS_T2_ values of the MLSS liquor of the MDBR tank in the range of 1214–3205 mg/L was observed due to the recirculation of MD reject back to the MD feed tank, and therefore accumulation of dissolved solids in the tank (Figure 5). The TDS values of the MD permeate were in the range of 0–3 mg/L which gives removal efficiencies of TDS in the range of 99.75–99.96%.

For Experiment 2, the TDS_T1_ of the MBBMR effluent ranged from 931–1079 mg/L. However, a gradual increase in TDS_T2_ values in the MDBR tank was observed and ranged from 981–3033 mg/L due to recirculation of MD reject back to the MD feed as mentioned above, as shown in Figure 6. The TDS_P_ were in the range of 7–64 mg/L, the removal efficiency of TDS ranged between 93.7–99.3%. The TDS values of the MBBMR tank were in the higher range due to.

#### 3.1.2. TSS of MBBMR and MDBR Effluents

The values of TSS_0_ represent the mixed liquor suspended solids (MLSS) concentrations of the MBBMR tank, while TSS_T1_ values refer to the effluents of the MBBMR tank. Similarly, the BOD_5,0_ is meant for the mixed liquor of the MBBMR tank, and BOD_5,T1_ refers to the effluents of the MBBMR tank. The variations of these parameters are depicted in Figure 7 (Experiment 1) and Figure 8 (Experiment 2). During the experiment period, the TSS_0_ increased from 72 to 119.5 mg/L and the TSS_T1_ increased from 50 to 98.5 mg/L. The removal efficiency of MBBMR for TSS varied from around 45% at the beginning of the testing period to negative values at the end which indicated insufficient acclimatization of microorganisms in the tank. While under the normal and steady operation of MBBMR, it is expected to achieve stable TSS_T1_ values and high TSS removal efficiency [26], the TSS_T1_ values increased with the increase in TSS_0_ values. This indicates that the testing period of 3 weeks is not adequate to achieve the required biodegradation rate of biomass. Arabgol et al. [27] reported that five weeks of operation are required for full inoculation of MBBR, followed by another three weeks for reaching steady-state operation. In the case of BOD_5_, the removal efficiency varied from 89% to 95%. 

As already reported for experiment 1, TSS and BOD_5_ values were measured similarly for experiment 2 (Figure 8). In this case, the TSS of the MLSS liquor ranged from 2142–3237 mg/L and the TSS removal efficiency in experiment-2 ranged between 61–98%, while initial BOD_5_ ranged from 3.76–7.18 mg/L and BOD_5_ removal efficiency was around 47–98%.

#### 3.1.3. MDBR Permeate Flux

The MDBR is a combination of membrane distillation separation and wastewater biological treatment in one process unit operation. The temperature of the bioreactor was maintained around 49 °C by recirculating the MLSS liquor in a heat exchanger placed in a heating bath operating at 50 °C. In Experiment 1 and Experiment 2, the hot MLSS liquor is recirculated through the shell side of a side-stream direct contact MD module. On the cold side of the MD module (lumen side), the temperature of the permeate water was maintained at 19 °C by recirculating the permeate water in a heat exchanger placed in a cooling bath operating at around 15 °C. The permeate water flux rate is the key performance parameter of the hybrid system and depends mainly on the temperature gradient across the MD membrane which is the driving force of the process. The MD module inlet feed temperature (TH,0) varied between 46.1–47.7 °C, and the MD module outlet temperature (TH,1) varied between 45.8–47.1 °C. The inlet MD condensate temperature (TC,0) varied between 16.9–19.1 °C, and the outlet condensate (permeate) temperature (TC,1) varied from 20.29–22.6 °C.

A comparison of permeate flux (J_p_) obtained during Experiment 1 and Experiment 2 has been illustrated in Figure 9. The MD permeate flux was observed to reduce linearly, due to fouling of the membranes or pore blockage, and the TDS elevation in the MLSS liquor. The average values of permeate flux for Experiment 1 and Experiment 2 were 3.3 and 2.6 L/(m^2^h), respectively, while the average TSD of MLSS liquor for Experiment 1 and Experiment 2 were 1948 and 2291 mg/L, respectively (Table 3). Therefore, it can be calculated that the percentage reduction in J_p_ (21%) is almost correlated with the percentage increase in TDS (18%). Hence, it can be concluded that the TDS plays a crucial role in the MD permeate flux [24]. When TDS of the permeate water was increased from 1.9 mg/L in Experiment 1, when the TSD of MLSS liquor was 1948 mg/L, to 32 mg/L in Experiment 2, when the TDS of MLSS liquor was 2291 mg/L (Table 3). Membrane wetting is the only possible reason for the increase of permeate water TDS. Experiment 2 was carried out after Experiment 1 without changing the MD module, hence loss of membrane hydrophobicity, and subsequently, the occurrence of membrane wetting may be attributed to the number of days of use as well as the increase in MLSS liquor salinity [14,24].

### 3.2. Assessment of Hybrid MDBR Configuration Performance for Secondary Wastewater Purification (Experiment 3)

In experiment 3, a hydrophobic hollow fiber membrane module was submerged in the MDBR tank, where the shell side comes into direct contact with the hot MLSS liquor, while the cold permeate is recirculated in the lumen side of the MD module. The hydrodynamic regime in the outer boundary of the membranes is affected only by the movement of air bubbles, while the permeate water velocity inside the lumen is related to the permeate recirculation flow rate which was fixed at 2 L/min. As illustrated in Figure 10, the permeate flux, J_p_, value varied in the range of 1.47–2.91 L/(m^2^.h). The experiment was carried out for a period of 60 days. The flux was observed to decrease in a more rapid rate in the first 7 days of the experiment, where the normalized flux declined from 1 to 0.69, then the rate of decrease remains moderate for the remainder of the experiment, where the normalized flux declined from 0.69 to 0.56. As explained earlier, the reduction in flux is due to the synergy effect of two factors i.e., the effect of membrane fouling and the effect of increased concentration of the MLSS liquor.

Figure 11 shows the variations of TDS_T2_ and TDS_P_ observed during the purification of secondary wastewater using the MDBR system (Experiment 3). TDS_T2_ values ranged from 890–1914 mg/L, and TDS_P_ values were in the range of 19–167 mg/L. This increase in TDST2 is due to the rejection of the dissolved ions by the MD module which results in the accumulation of dissolved ions in the bioreactor tank. It should be noted that at Day 17 when the TDS_T2_ concentration reached 1705 mg/L, the MDBR tank was emptied and refilled again with feed secondary treated wastewater. In contradiction to the observed trend of correlation between TDS_T2_ and TDS_P_ (Figure 6), the TDS_P_ trend in Experiment 3 initially increased in the first six days of operation, then decreased to an almost stable level at Day 17 till the end of the experiment. Although this can indicate the stable hydrophobic characteristics of membranes, it can be also related to the bubbling effect on the hydrodynamic regime in the outer boundary of the membranes and the fouling mitigation effect of bubblers. 

### 3.3. Contact Angle Measurement

Contact angle (θ) is an essential parameter for evaluating a membrane’s hydrophilicity and wetting behavior [28]. The contact angle of a surface is measured by placing a drop of liquid and measuring the angle formed between the surface and the line tangent to the edge of the drop of liquid. A low contact angle indicates high surface energy and, therefore, a high hydrophilic character of the membrane. Vice versa in the case of high contact angle. Contact angle provides quantitative data about the wettability of a surface at a molecular level [28]. The measure of the contact angle allows for assessing a material surface’s quality before an adhesion process. The same sidestream MD membrane module was used for Experiment 1 and Experiment 2. After completion of Experiment 2, the MD module was disassembled, and three fibers were cut and used for the contact angle measurement. The measured values of the contact angle are shown in Table 4, where it can be seen that the average values were ranged from 60.1° to 63.2°. When compared to the pristine fiber, the contact angle of the used fiber worsened by 44.7%, which indicates the possibility of hydrophobicity loss. However, the rate of hydrophobicity loss with time is low compared to other MD membranes used for wastewater treatment [17]. Conventional hydrophobic MD membranes (i.e., membranes that display apparent contact angle θ* > 90° with high surface tension liquids such as water) suffer from membrane wetting in desalination of feedwater containing low surface energy contaminants (e.g., shale gas-produced water [29,30] and coal seam gas produced water [31].

The loss of hydrophobicity and the concomitant decrease in the contact angle of the membrane progresses with the time of operation [32]. The accumulation of foulants on the membrane surface, and in the membrane pores is a time-dependent process and is the main factor for hydrophobicity degradation [33]. The contact angle of the membrane can be measured only if the membrane module is disassembled. Therefore, the practical method for assessing the changes in contact angle is by observing the trend of change of salinity and flux rate of MD permeate i.e., TDS_p_ and J_p_ values. In Experiment 1, where a new MD membrane module was used, the TDS_p_ values were stable around an average value of 1.7 mg/L with no significant increase with time (Figure 5). However, in Experiment 2, where the same MD membrane module was used after cleaning with deionized water, the TDS_p_ values maintained approximately constant level in the nine days of operation at an average of 14 mg/L, then start to increase progressively up to 75 mg/L after 16 days of operation. The manner of J_p_ change with time of operation is another indicator of contact angle change. In experiment 1, the permeate flux reduction of 20% was observed after 14 days of operation, while in Experiment 2, the permeate flux reduction of 35% was observed after 16 days of operation.

After the completion of Experiment 3, the contact angle of the submerged MD membrane was measured (Table 5), and the average value was found as 84.75–85.4°.

By comparing the average contact angle values for the side stream module and submerged module, it can be deduced that the hydrophobicity loss of the submerged module is lower. As noted by Morrow et al. [34] in sidestream configuration, fouling is mitigated with hydraulic crossflow; however, in submerged configuration, the fouling is mitigated via air scour. 

### 3.4. Comparison of Performance Parameters of Different System Configurations

In experiment 1, the TDS_P_ values of the MD permeate ranged from 1–3 mg/L. Whereas in experiment 2, the TDS_P_ ranged from 8–64 mg/L, and in experiment 3, the TDS_P_ ranged from 25–167 mg/L. A higher degree of treatment was obtained in experiment 1 which had secondary treated water as the feed water. Whereas in experiment 2, when primary treated water was used as feed water, the worse values for TDS_P_ and J_P_ were obtained. Table 3 provides a summary of experimental results indicating averages of performance parameters of the proposed configurations for industrial wastewater treatment.

Membranes used in Experiment 1 were cleaned with distilled water for a period of 30 min before being used in Experiment 2. Since secondary treated water was used in Experiment 1, and the membrane was new, the permeate showed the best results. In Experiment 2, primary treated water was used together with the membrane used in experiment 1 and showed worse results comparatively. 

The flux decreased in all three experiments, with a higher flux in Experiment 1. In experiment 2, the TDS decreased along with the permeate flux until day 8. After this, an increase in TDS was noticed due to the clogging of pores compared to initial conditions. In Experiment 3, the TDS and flux increased during day 2 and then decreased; during this time, there was no change in pressure or temperature of the influent on the system.

### 3.5. Comparison of the Quality of the Water Produced by the Three Different Configurations with the Quality of the Water Produced by Jeddah Industrial Wastewater Treatment Plant

To compare the treatment efficiency achieved with the proposed configurations in this study, water quality analysis data reports were collected from Jeddah Industrial City Wastewater Treatment Plant, operated by Modon (JICWTP). In this plant, around 40% of the secondary effluents are purified by the advanced treatment process consisting of a sand filter, UF, and RO. The comparison points were chosen at the UF product outlet and the RO product outlet. The daily water analysis reports of the advanced treatment process were collected for fifteen days, and the average values of TDS of the RO product water, and TSS of the UF product water were calculated for comparison. 

The TDS values for the RO effluent at the JICWTP and MD permeate for all three experiments are shown in Figure 12a. As expected, the average TDS value of the MD product for the Exp1 configuration in which secondary treated wastewater was used as a feed for the MBBMR step was less than 2 mg/L. However, the average TDS of the advanced RO treatment product was 106 mg/L. When primary wastewater was used as feed for the MBBMR step, the TDS of the MD product increased to around 24 mg/L probably due to the wetting of the MD membrane by surfactants and oil residues present in the primary feed water. Further, the TDS of the MD permeate was worsened in the case of the Experiment 3 configuration in which the MBBMR step was excluded. Further, The TSS values of UF effluent from the JICWTP are compared to the MBBMR setup in Experiment 1 and Experiment 2, as shown in Figure 12b. The comparison of the TSS data showed higher performance of the advanced RO treatment at JICWTP compared to the configurations tested in this work. The use of a sand filter prior to the UF step eliminates oil and nanoparticles present in the secondary wastewater by absorption mechanism, resulting in a TSS value of 1 mg/L. However, in the case of Experiment 1 and Experiment 2 configurations, the MBBMR treatment is not efficient for the removal of these contaminants. Therefore, the TSS values of 76 mg/L and 110 mg/L were found.

## 4. Conclusions

Membrane distillation is a potential high rejection, the terminal process for onsite industrial wastewater treatment when waste heat is available. Membrane wettability threat is the main challenge for MD application in industrial wastewater which contains high concentrations of dissolved organic matter and surfactants. This work compares the role of pretreatment steps prior to MD on the MD performance efficiency. A new integrated membrane system configuration consisting of MBBR integrated with UF, followed by MBR integrated with MD, was tested for treatment and purification of primary and secondary wastewater treatment. This configuration was compared with a simple integrated membrane system configuration consisting of MBR integrated with MD only. The MD performance efficiency achieved by the first configuration was better than that achieved by the second configuration. Additionally, the results proved that the quality of MD permeate remains stable when applying the 3-step process. A correlation has been found between TSS in the MD feed and the TDS of MD permeate. Hence, TSS should be maintained low by introducing a sand filter for absorbing contaminants that are still present after the MBBMR step. 

## Figures and Tables

**Figure 1 membranes-13-00016-f001:**
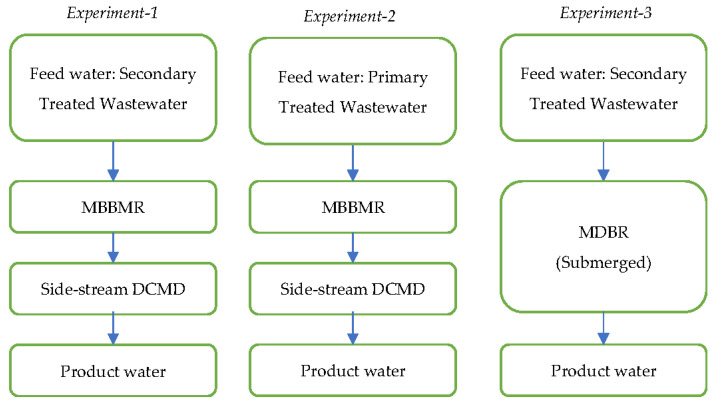
Configurations of the hybrid wastewater treatment and membrane processes tested, and the related experiments.

**Figure 2 membranes-13-00016-f002:**
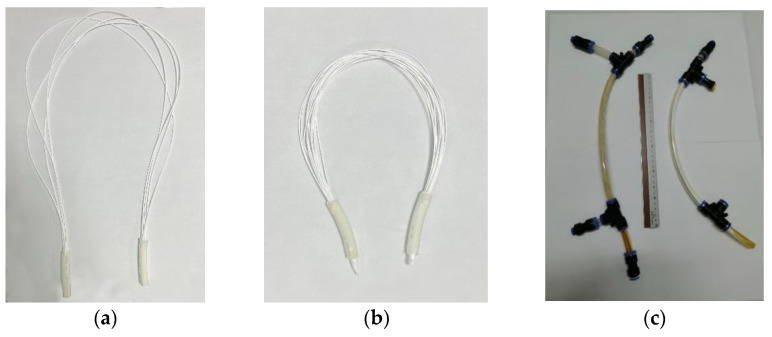
Experimental membrane modules used in the study; (**a**) UF membrane module used in hybrid MBBMR configuration, (**b**) MD membrane module used for submerged MD (Experiment 3), and (**c**) MD membrane module used for side stream MD (Experiment 1 and Experiment 2).

**Figure 3 membranes-13-00016-f003:**
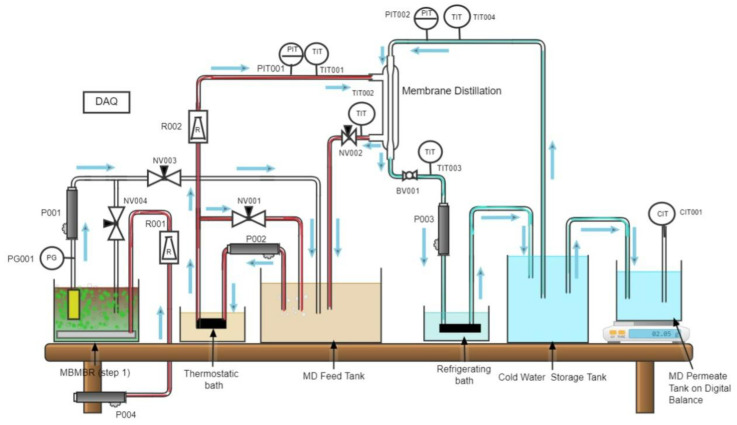
Schematic of the hybrid MBBMR and side stream DCMD configuration used in Experiments 1 and 2. (PG00x: pressure gauge; P00x: pump; NV00x: needle valve; R001: air flow meter, R002: water flow meter; PIT00x: pressure transducer; TIT00x:. thermocouples; CIT001: conductivity transmitter. DAQ: data acquisition).

**Figure 4 membranes-13-00016-f004:**
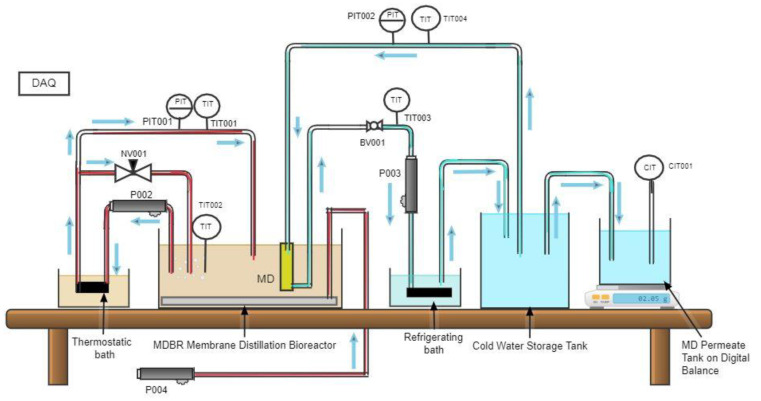
Schematic of the hybrid submerged MDBR configuration 2 used in Experiment 3. (PG00x: pressure gauge; P00x: pump; NV00x: needle valve; R001: air flow meter, R002: water flow meter; PIT00x: pressure transducer; TIT00x:. thermocouples; CIT001: conductivity transmitter. DAQ: data acquisition).

**Figure 5 membranes-13-00016-f005:**
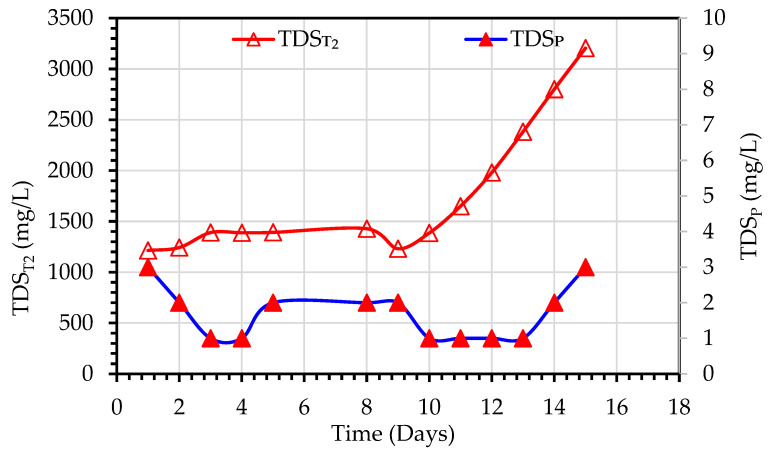
Variations of TDS values observed during the treatment of secondary wastewater using a hybrid MBBMR and DCMD system (Experiment 1).

**Figure 6 membranes-13-00016-f006:**
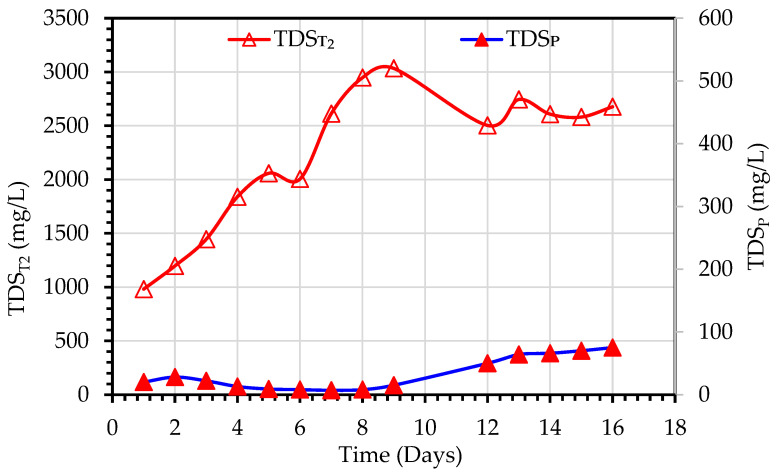
Variations of TDS values observed during the treatment of primary wastewater using a hybrid MBBMR and DCMD system (Experiment 2).

**Figure 7 membranes-13-00016-f007:**
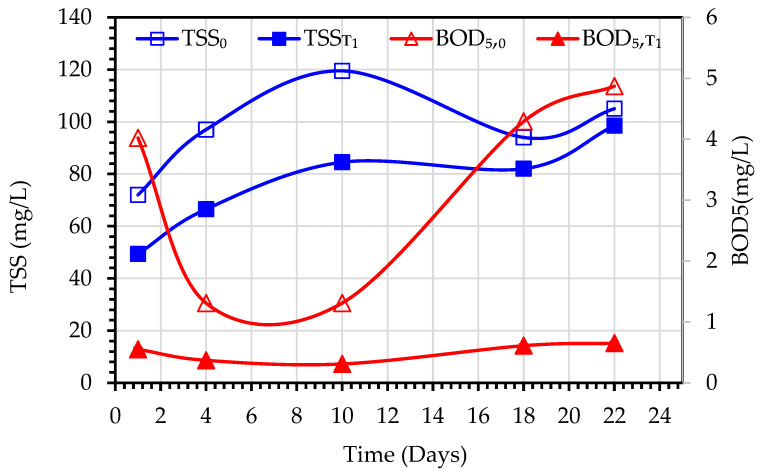
Variations of BOD5 and TSS values observed during the treatment of secondary wastewater using a hybrid MBBMR and DCMD system (Experiment 1).

**Figure 8 membranes-13-00016-f008:**
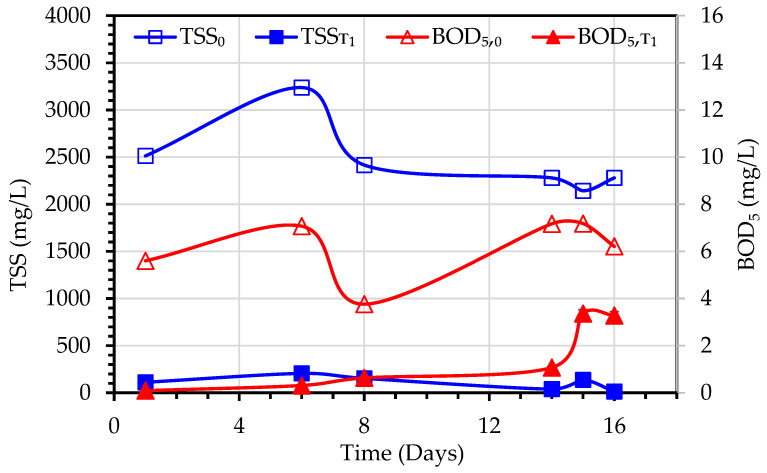
Variations of BOD_5_ and TSS values observed during the treatment of primary wastewater using a hybrid MBBMR and DCMD system (Experiment 2).

**Figure 9 membranes-13-00016-f009:**
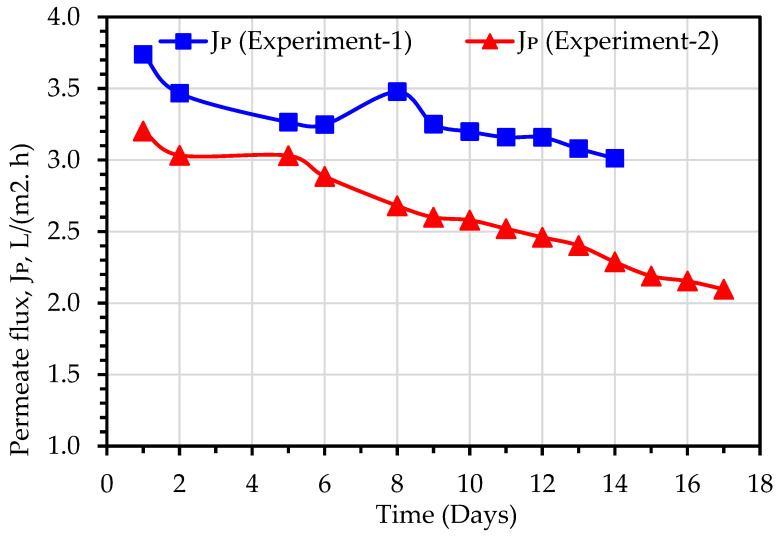
Comparison of MD permeate flux values observed during purification of secondary wastewater (Experiment 1) and primary wastewater (Experiment 2) using hybrid MBBMR and DCMD system.

**Figure 10 membranes-13-00016-f010:**
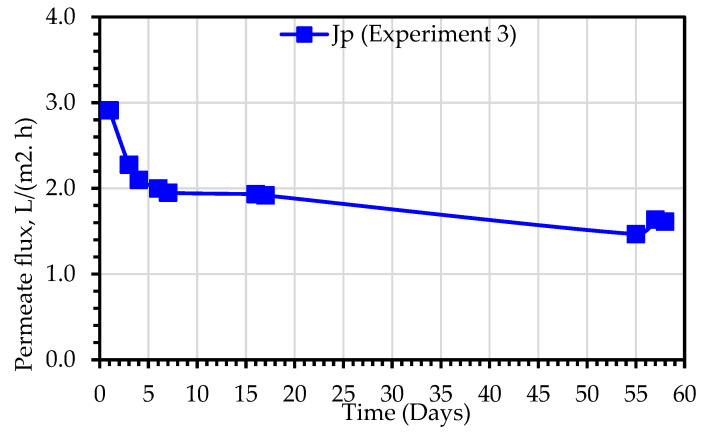
Permeate flux observed during treatment of primary wastewater using a hybrid DCMD system (Experiment 3).

**Figure 11 membranes-13-00016-f011:**
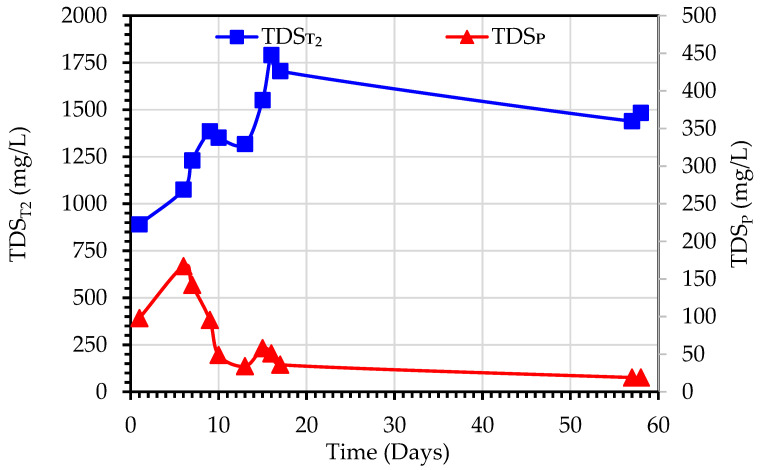
Variations of TDS values observed during the treatment of secondary wastewater using a hybrid DCMD system (Experiment 3).

**Figure 12 membranes-13-00016-f012:**
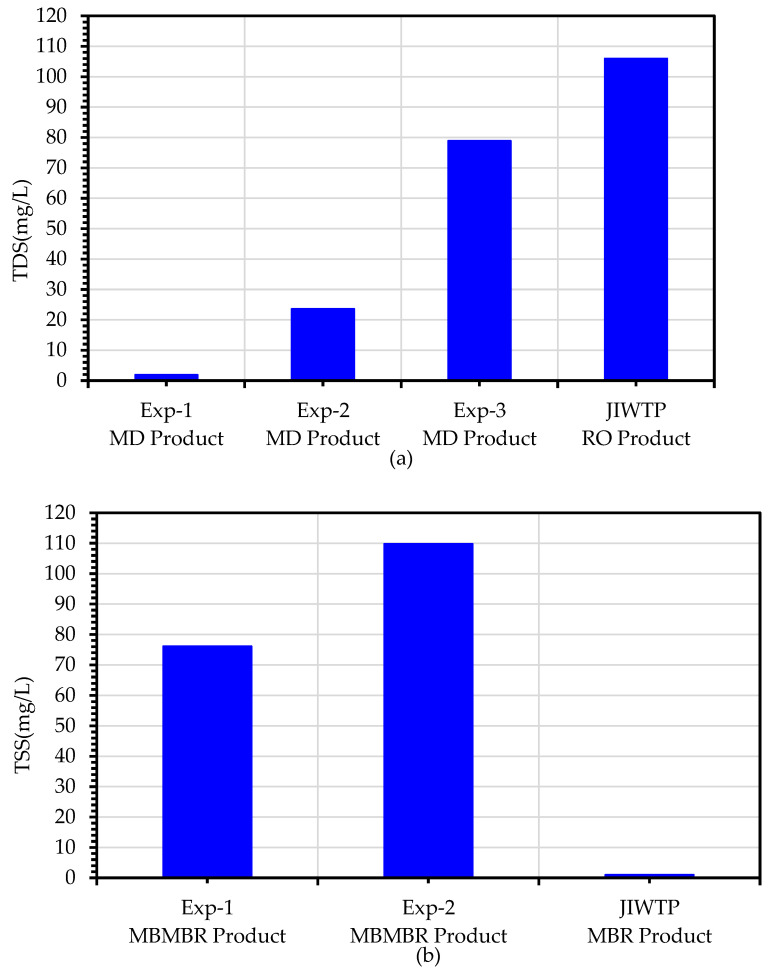
Comparison of TDS and TSS values obtained during treatment of secondary treated wastewater using hybrid MBBMR and DCMD system (Experiment 1 and Experiment 2) with respective values reported by Jeddah Industrial Wastewater Treatment Plant: (**a**) TDS (**b**) TSS.

**Table 1 membranes-13-00016-t001:** Properties of UF modules used in the study.

Properties	Units	UF Membrane
No of membranes in a module	number	4
Length	cm	80
Contact angle [°]	[°]	85
Outer diameter	mm	2.20
Wall thickness	mm	1
Porosity	%	55.6
Mean pore size	nm	340–390
Maximum Load	N	333
Tensile Strength	N/mm^2^	87.7
Elongation	%	25.8
Modulus (Automatic Young’s)	MPa	612.5
Load at 1%	N	10.4
Tensile stress at 0.2%	N/mm^2^	87.4
Membrane area	m^2^	0.02212

**Table 2 membranes-13-00016-t002:** Properties of MD modules used in the study.

Properties	Side-Stream MD Module	Submerged DCMD Module
Membrane material	PVDF	PVDF
Mean pore size (μm)	0.2	0.2
Number of hollow fibers	15	13
Nominal inner diameter of the fiber (mm)	0.80	0.80
Nominal outer diameter of the fiber (mm)	1.2	1.2
Effective membrane area (m^2^)	0.0113	0.00139
Effective module length (m)	0.30	0.37
Effective module’s membrane area (m^2^)	0.01696	0.01813

**Table 3 membranes-13-00016-t003:** Summary of experimental results indicating averages of performance parameters of the proposed configurations for industrial wastewater treatment.

Duration of Experiment (Days)	Parameters (Pi)	Feed Wastewater (i = f)	MBBMR Tank (i = 0)	MBBMR Filtrate (i = 1)	MD Tank (i = 2)	MD Permeate (i = p)
Experiment 1						
27	TDS_i_ (mg/L)	818.0	945.6	770.5	1947.8	1.9
	pHi	9.4	7.9	8.8	8.0	6.4
	TURi (NTU)	1.6	0.6	0.6	1.6	-
	BOD₅,i (mg/L)	6.0	2.5	1.0	-	-
	TSSi (mg/L)	89.8	97.5	82.9	-	-
	TOCi (mg/L)	11.7	11.5	12.0	-	-
	Ti, °C	-	22.1	20.2	47	19
	Jf (L/m².h)	-	44.23	-	-	-
	TMP (psi)	-	−6.1	-	-	-
	J_p_ (L/(m².h)	-	-	-	-	3.3
Experiment 2						
16	TDSi (mg/L)	962.0	1037.4	922.7	2291.3	32.0
	pHi	4.5	6.8	7.2	7.6	6.7
	TURi (NTU)	40.0	1575.4	1.5	4.5	-
	BOD₅,i (mg/L)	-	5.1	3.3	-	-
	TSSi (mg/L)	524.1	1643.4	151.6	-	-
	TOCi (mg/L)	286.3	44.8	28.5	-	-
	Ti, °C	-	19.8	19.8	47	18
	Jf (L/m².h)	-	31.6	-	-	-
	TMP (psi)	-	−7.0	-	-	-
	J_p_ (L/(m².h)	-	-	-	-	2.6
Experiment 3						
74	TDSi (mg/L)	818.0	-	-	1729.4	75.0
	pHi	9.4	-	-	8.3	7.1
	TURi (NTU)	1.6	-	-	0.9	0.3
	BOD₅,i (mg/L)	6.0	-	-	-	2.0
	TSSi (mg/L)	89.8	-	-	577.5	-
	TOCi (mg/L)	11.7	-	-	21.7	
	Ti, °C	-	-	-	46	19-
	J_p_ (L/(m².h)	-	-	-	-	2.182

**Table 4 membranes-13-00016-t004:** Contact angle results of membranes used in Experiment 1 and 2.

Membrane	Pristine Fiber	Fiber 1 (Used)	Fiber 2 (Used)	Fiber 3 (Used)
Location 1	109.4	64	66	58.9
Location 2	112.5	62.4	55.4	61.3
Average	111	63.2	60.7	60.1

**Table 5 membranes-13-00016-t005:** Contact angle results of membranes used in Experiment 3.

Membrane	Pristine Fiber	Fiber 1 (Used)	Fiber 2 (Used)	Fiber 3 (Used)
Location 1	109.4	84.4	85.2	84.9
Location 2	112.5	85.3	85.6	84.6
Average	111	84.85	85.4	84.75

## Data Availability

Not applicable.

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
