# Peer review of "Evaluation of a Hybrid Moving Bed Biofilm Membrane Bioreactor and a Direct Contact Membrane Distillation System for Purification of Industrial Wastewater"

_membranes, 2022, doi:10.3390/membranes13010016_

Round 1

Reviewer 1 Report

The manuscript presents a study on the integration of MBMBR and MD technologies for industrial wastewater treatment. Both configurations are quite innovative and therefore their evaluation especially using real wastewater is of interest. Still I have some concerns about the manuscript in its current form that should be largely improved before potential consideration for publication:

My main concerns are:

·       Overall, the manuscript is very scholar with a succession of information but a lack of critical thinking/ interpretation discussion

·       The introduction remains like a succession of the background and somehow state of the art but the authos should better focus and the closets state of the art/interest and limitations of tested configurations to justify better their study and its novelty

·       One crucial point is that in abstract authors talk about MBR, MBBR and then in the manuscript of MBMBR…lack of coherence…If MBMBR configuration is novel and not just the succession of MBR and MBBR then the authors should emphasize on it and if it has already been tested, mention the closest state of the art

·       Figures are very basic (typical basic excel figures…) and should be improved. Moreover, some figure presented like pH are of very limited interest

·       There is very limited/no discussion on the observations in the R&D section. The author should discuss their results, refer to literature, bring hypothesis, etc…

Some more specific comments:

·       Many tipos: L119: A?, L120: a.? L140: 5 L.m2.h?.....

·       Feed water characteristics are missing

·       L225-227: abbreviation are not clear

·       Fig 6: whsy such changes in BOD0??

·       3.4: contact angle intro should be in mat&met

·       Change in contact angle could also be due to different operation time? More loss of hydrophobicity is more operation time? Also is loss of hydrophobicity due to change in memebrane materials or fouling/scaling? That should be discussed

·       L382: data on the performance of sand filtration effcicency? Water quality?

Author Response

Dear Revewier 

Dear Reviewer,

Thank you very much for your valued comments. Attached file has the responses for the comments and they have been taken in consideration for the updated manuscript.

Reviewer 2 Report

The authors present very interesting project but some questions and comments are necessary, as well as some corrections (see also the pdf version of the manuscript)

Knowing that MD is not energetically more efficient than reverse osmosis, why MD was used in this work as third step for water purification?

The manuscript is descriptive. The results have to be discussed comparing used processed and with the results of other authors. I suggest to include one resume table comparing the efficiency of studied parameters for the three studied processes.

Line 155: to specify the nature of industrial waste water

Lines 158-160: The specifications of primary and secondary treated waste waters have to precise in a table or supplementary information.

Lines 158-160: Why used proposed system after secondary WWT and not only after primary WWT?

Line 240: To precise in Fig. 5 the mean of Ph0, pHT1, pHT2, PHP, …TDSP. Idem in other figures when necessary.

Line 246: in the axes of Fig. 6 to check the writing of concentration units: mg/L. To check in the rest of the manuscript.

Line 292, Fig. 10: to check units in the axes

In the experimental section the authors, describing experimental setups, mention the presence of the gauge for TMP measurement. However, not any result about the variation of TMP is presented. This parameter is very important and a critical one when using membranes. I suggest to present these results and discuss them. How to deal with the cleaning/changing of membranes in the studied processes?

I suggest to include the conclusions of the presented work.

Author Response

Dear Reviewer,

Thank you very much for your valued comments. Attached file has the responses for the comments and they have been taken in consideration for the updated manuscript.

Reviewer 3 Report

The paper is hard to read. There is much information that probably could be omitted. I am not sure if all results are fascinating.

I would like to know the cut-off or pore diameter of the membranes.

The authors forgot to present any conclusions.

Author Response

(The authors gave the same response as above.)

Round 2

Reviewer 1 Report

The manuscript presents a study on the integration of MBMBR and MD technologies for industrial wastewater treatment. Both configurations are quite innovative and therefore their evaluation especially using real wastewater is of interest. The manuscript has been improved significantly with regards to the previous version. I still have the following comments that should be addressed before being considered for publication

My main concerns are:

·       Still to be improved: The introduction remains like a succession of the background and somehow state of the art but the authors should better focus and the closets state of the art/interest and limitations of tested configurations to justify better their study and its novelty. Be more synthetic and convincing to demonstrate the interest of your study

·       There is still limited discussion on the observations in the R&D section. The author should discuss their results (after the figures) and refer to literature, bring hypothesis, etc…

Author Response

A major revision of the structure of the Results and Discussion Section has been made, without changing contents which have been already revised by the respected reviewers. Also, more in-depth discussion has been added to this section.   

Reviewer 2 Report

Thank for the corrected manuscript

Author Response

Thank you for your comments.

Reviewer 3 Report

I accept the present version of manuscript

Author Response

Thank you for your comments.

Round 3
